# Exploring International Faculty's Perspectives on Their Campus Life by PLS-SEM

Chia-Chi Chen [1] and Dian-Fu Chang [2,*]

1   Language Center, Shih Chien University, Taipei City 104336, Taiwan; sophiabv03@gmail.com
2   Department of Education and Futures Design, Tamkang University, New Taipei City 251301, Taiwan
*   Correspondence: 140626@mail.tku.edu.tw

**Abstract:** The purpose of this study was to explore the perspectives of international faculty members on their life in higher education institutions (HEIs). The COVID-19 pandemic has impacted not only most citizens' lives but also the international faculty members' lives during this period. Since building sustainable campuses has become a priority for various HEIs, attracting and maintaining international faculty have become the focus of various internationalized campuses. However, the issue of international faculty's satisfaction is still neglected in higher education. Using a self-compiled online survey, we collected 80 international faculty members of HEIs in Taiwan to investigate this issue. About 31.25% of the responses were collected by the online survey technique. The survey covered the career and professional status, teaching and research status, and demographics of the faculty. This study proposed a novel conceptual framework for addressing international faculty's campus life, the design of which examined the relationships among working conditions, views of institutions, views of government measures, and levels of satisfaction through partial least squares structural equation modeling (PLS-SEM). The demographic profile of the participants revealed that (a) most international faculty are employed full-time as lecturers or assistant professors in most public universities and (b) most international faculty members earned their first degree outside of Taiwan; however, 66% of them earned their post-doctoral degrees in Taiwan. The result of the PLS-SEM confirms that the international faculty's perspective of government, through their current work satisfaction, impacts overall satisfaction. This study found a mediation effect in the testing model. The design of the study can be extended to other higher education settings to tackle similar issues.

**Keywords:** internationalization; international faculty; higher education; job satisfaction; PLS-SEM

## 1. Introduction

In higher education settings, international mobility may refer to the movement of both students and faculty across countries. There are numerous studies that addressed international student mobility in higher education settings, while studies focusing on international faculty mobility are still limited, inconsistent, and incomplete [1]. For example, international faculty studies are mostly based on qualitative approaches in specific countries. Tan indicated there is no international database about international faculty trends available at the current stage [2]. International faculty recruitment has the potential to allow for a better understanding of its positive and negative consequences with profound data collection and deeply rethink the competitive recruit-and-retain policy in the global context. Therefore, a novel approach to investigate this issue for sustainable higher education is needed.

In the global context, for example, the Korean government launched the "Brain Pool Project" to attract international researchers in 1994; Japan has made greater efforts to attract international faculty, researchers, and talents by implementing several national-level policies in the 1990s. Attracting international faculty has become an increasingly important strategy for improving the quality and international competitiveness of national

higher education systems in the OECD areas and many East Asian countries such as China, Singapore, and Malaysia. In Taiwan, the Ministry of Education focused on selected higher education institutions possessing a reputation for high-quality research and launched the "Development Plan for World Class Universities and Research Centers of Excellence" in 2016. In the second stage (2011–2016), the project was renamed the "Aim for the Top University Plan". Among these initiatives, international faculty recruitment has since been encouraged at an institutional level. Facing the pressure of international academic mobility, the Ministry of Education initiated a project called the Higher Education Sprout Project from 2018 to 2022. Under this program, universities can recruit the world's elite by offering annual salaries of up to TWD 5 million (about USD 167,000) [3]. However, the recruitment of outstanding international scholars is still limited.

International mobility involves leveraging the knowledge-producing skills of select individuals in exchange for highly attractive living and working conditions. In general, academic mobility is not a singular or universal process but rather is multi-faceted and better understood when examined in light of particular circumstances [4]. Numerous universities are now seen to be central in the global competition for knowledge, innovation, and human capital. In this sense, effective recruitment and retainment of international talents may become an important strategy for international campuses. However, the situation varies by country and area; for example, developing and developed countries may face different situations. The phenomena of brain drain, brain gain, or brain circulation have often guided inquiry into international academic mobility. The purpose of this research was to explore the working conditions and perspectives of international faculty in higher education by using innovative approaches.

Taking Taiwan's higher education as an example, this study identified the current working conditions of the international faculty to determine the influential factors that might attract more international faculty members to Taiwan. Partial least squares structural equation modeling (PLS-SEM) is an analytical approach to deal with composite-based and causal–predictive models, and it has been proposed to tackle structural issues in different disciplines. When we are confronted with new issues, PLS-SEM is well suited for exploratory research [5]. An increasing number of higher education studies use the PLS-SEM method for explanations and predictions [6]. For practical reasons, PLS-SEM can accept and handle small samples when the research subjects are limited [7]. Considering the constrained population of international faculty, the proposed study could focus on PLS-SEM to demonstrate that the data were transformed and the research hypotheses were verified. Specifically, there were three major purposes in this study: (a) to examine the problems that the current international faculty faced in higher education institutions through a survey, including their career and professional situation and teaching and research situation; (b) to determine the international faculty's perspective on current working conditions, including their institutional view, governance view, current work satisfaction, and overall work satisfaction, in order to examine the relationships among these factors; and (c) to provide some suggestions for retaining international faculty and attracting newcomers. Based on the research purposes, we addressed the following research questions:

a.  What are the problems facing international faculty work at national and institutional levels?
b.  What are the international faculty's perspectives on the current working environment?
c.  What are the relationships among international faculty's institutional view, governance view, current work satisfaction, and overall work satisfaction?
d.  Which strategy can be used to ameliorate the working conditions for international faculty?

The remaining parts of this paper are organized as follows: First, the literature review addresses the notions of international faculty mobility, international faculty in Taiwan, and related research addressed in previous studies which can support this study. Second, the method section displays the research framework, hypotheses, instrument, samples, and statistical analysis. Third, the results section demonstrates our descriptive statistics,



measurement model, and the testing of the structural model. Fourth, the discussion is presented. Finally, the conclusions are drawn.

## 2. Literature Review

In this section, first, we focus on the definition of international faculty. Second, we address the phenomenon of international faculty mobility, its meaning for international higher education, and the target higher education system and related internationalized policies. Third, the related approaches are reviewed. Fourth, we address job satisfaction and academic satisfaction as the main theme to develop the survey questionnaire. Finally, we address the research hypotheses.

### 2.1. Definitions of International Faculty

The definition of international faculty is complicated and varied. After the establishment of modern nation states in the early nineteenth century, the term "international faculty" became widely used, and the profession of academia was created [8]. Mihut et al. pointed out that there is no generally agreed-upon answer to the question of what it means to be "international" because of challenges associated with the diversity of motives, lengths of stay, and modes of mobility among this population [9]. Intrinsically, most research on international faculty with border classification is by foreign-born (place of birth) or by non-citizens (citizenship) [10]. Altbach and Yudkevich defined international faculty as individuals who hold academic positions in countries in which they were not born or in which they did not complete their first post-secondary education [11]. In statistical research, for instance, the Quacquarelli Symonds (QS) World University Ranking methodology defined international faculty as simply based on the proportion of faculty members that are international in order to build a composite indicator as QS scores [12]. It goes without saying that definitions of international faculty can be narrow based on different types of categories that are based on different countries and research purposes; however, many studies on international faculty use nationality as the definition for the research, especially in Asian countries [8,13]. The literature review suggests that international faculty members are classified according to their nationality.

### 2.2. International Faculty Mobility

2.2.1. Internationalization as Main Driving Factor of International Faculty Mobility

One of the key issues in the situation of economic globalization is the pressure of enhancing internationalization in higher education [14,15]. International student and faculty mobility has become the key international indicator. The internationalization of higher education has made the mobility of students and faculty the main recruiting and retaining strategy for academic reasons [16]. The OECD claimed the importance of mobility stems from its contribution to the creation and diffusion of knowledge; similarly, the Global Education Monitoring Report investigated the shifting mobility in international higher education [17,18]. Moreover, Bhandari et al. reported that moving educational programs beyond student and faculty mobility can contribute to the flow of ideas and knowledge, improving practices, generating resources for countries receiving them, and attracting talents [18]. Within the academic mobility context, mass higher education has accelerated the process of transformation from higher education importers to exporters. This is due to higher education internationalization having directly impacted international faculty mobility. Hudzik, who conceptualized internationalization in HEIs, indicated that this phenomenon may include internationalized curricula and the hiring of more international faculty members [19]. Numerous studies suggested that internationalization should be taken into account as the main target and considered international faculty mobility in relation to the specific phenomenon of internationalization of higher education [11,20,21]. Internationalization has become a main driving factor that impacts international faculty mobility.

### 2.2.2. International Faculty in Taiwan

Taiwan has established two forms of higher education: academics and occupational training. A total of 152 colleges and universities are currently operating in Taiwan's HEIs, including 126 universities, 14 colleges, and 12 junior colleges [22]. Taiwan's international­ization policy could be viewed as focusing on the mobility of international students [23], while how to recruit and retain international faculty has persisted in discussion and catches much more attention in higher education. The MOE database revealed that the number of full-time foreign faculty members in higher education had a limited increase from 2017 to 2020. The average international faculty growth is less than 1% in Taiwan. According to the structures of international faculty in higher education, there were 1,170 international faculty members in HEIs in 2021. By percentage of international faculty members, 52.56% of faculty members are employed by public HEIs, while 47.43% are employed by private HEIs [24].

To enhance institutional internationalization, the government has initiated several measures in the past decades. For example, Taiwan's government revised the University Law in order to strengthen universities' autonomy to develop academic exchanges and partnerships with foreign cultural and educational institutions in 1994 [25]. Since then, the White Paper on University Education, "Ten Educational Development Policies", was initiated in 2001, and several universities have been referred to as active participants in internationalization activities [23,26]. The other influential factors considered in evaluating Taiwan's competitiveness include student mobility, rankings, and employment of interna­tional faculty members [27]. From that time forward, HEIs have been encouraged by the Ministry of Education to develop each institution's unique characteristics under a variety of incentive programs. In the 2000s, the MOE implemented two major incentive projects to promote the diversification and classification of higher education [28]. The first major incentive project is the Top University Project, which aims to increase the quality of research and ensure the inclusion of top global universities from 2006 to 2015. The second one is the Higher Education Sprout Project, which is scheduled to take place between 2018 and 2022 for the purpose of promoting the development of diversified higher education [29]. Within this context, internationalization has become an influential indicator for evaluating universities' performance. International faculty could play an important role in the process of institutional internationalization. Policy makers assumed that the expected progress could enhance higher education's sustainable development.

### 2.3. Approaches for Realizing International Faculty

Based on previous studies, research approaches for international faculty have been examined by empirical investigations for many years. For example, considering that inter­national faculty have individual experiences, Omiteru et al., based on demographic infor­mation, measured their perceptions about administrators and respective communities [30]; Huang used the demographic profile of international faculty to analyze their personal, edu­cational, and professional characteristics in Japan [31]. Similarly, using survey questions, Huang et al. sought to understand Japan's academic market for international faculty and institutional climate [32]. The number of international faculty was also considered one of the critical factors for promoting teaching variety and quality in previous studies [21,22]. However, quantitative studies could have a deeper analysis of the internationalization of the faculty studies along with the complex characteristics at the institutional levels. In the other direction, Munene adopted the embedded intergroup theory to pay attention to members of organizations [33]. Lawrence et al., based on the organizational equilibrium theory, identified "pull" and "push" variables of uncertain faculty leaning toward leaving or staying in their current institutions [34,35]. In addition, mixed methods were used to tackle this issue. For example, Huang conducted semi-structured interviews and surveys in order to collect quantitative data about the characteristics and motivations of foreign faculty members working in Japanese universities [15]. Similarly, Kim et al. used the concept of the push-and-pull model to seek the mobility patterns of foreign-born faculty [34,36].

This format of study also adopted the push–pull model to analyze international mobility with qualitative data analysis [15,37]. Push and pull factors may provide some reasonable interpretation of the international faculty's life and satisfaction.

### 2.4. Job Satisfaction and Academic Satisfaction

Job satisfaction could be one of the crucial indicators in understanding international faculty's campus life. For example, Hagedorn addressed international faculty perspectives on job satisfaction [38]; Mamiseishvili and Lee applied a theoretical model of faculty job satisfaction [39]; Nyquist et al. developed a model linking organizational, job-related, and individual factors to help evaluate faculty job satisfaction [40]. As Nyquist et al. noted, variables such as organization factors, job-related factors, and personal factors were viewed as outcomes of self-knowledge or social-knowledge satisfaction [40]. Self-knowledge satisfaction or social-knowledge satisfaction can be a trigger that could influence outcome productivity, retention satisfaction, or intrinsic rewards. Academic satisfaction may refer to specific content on campus. In the case of the Malaysian academic community, Rahman et al. confirmed that work-to-family conflict, family-to-work conflict, and work–family balance are the predictors of job satisfaction. Their findings reveal that work-to-family conflict and family-to-work conflict have negative significant effects on job satisfaction [41]. Previous studies indicated that customer satisfaction is the crucial quality assurance component of TQM; for example, TQM practices are significantly and positively linked to customer satisfaction and service quality [42,43]. In this sense, the notion of TQM and practices can be extended to realize international faculty in higher education. Job and academic satisfaction could be useful indicators to reflect the related policies, strategies, and campus life.

### 2.5. Hypotheses

In this sense, the questionnaire about satisfaction might include more details about academic-related activities. This study developed a structural relationship model to interpret international faculty's perspectives on satisfaction. We considered general job and academic satisfaction as an expected outcome in the research framework. Based on the previous discussion, we list the research hypotheses as follows:

**Hypothesis 1 (H1).** *International faculty's institutional view (IV) influences their current work satisfaction (CWSat).*

**Hypothesis 2 (H2).** *International faculty's governance view (GV) influences their current work satisfaction (CWSat).*

**Hypothesis 3 (H3).** *International faculty's current work satisfaction (CWSat) influences their overall satisfaction (OSat).*

**Hypothesis 4 (H4).** *International faculty's institutional view (IV) influences their overall satisfaction (OSat).*

**Hypothesis 5 (H5).** *International faculty's governance view (GV) influences their overall satisfaction (OSat).*

**Hypothesis 6 (H6).** *International faculty's institutional view (IV), through current work satisfaction (CWSat), influences their overall satisfaction (OSat).*

**Hypothesis 7 (H7).** *International faculty's governance view (GV), through current work satisfaction (CWSat), influences their overall satisfaction (OSat).*

## 3. Method

This study employed quantitative approaches to explore international faculty's career, professional situation, and their perspectives on HEIs. Typically, structural equation modeling (SEM) is applied to analyze data and verify hypotheses about interactions between international faculty and colleagues, departmental climate, and recognitions. The two most prevalent SEM-based analytical methods are CB-SEM (covariance-based SEM) and PLS-SEM (variance-based SEM) [44]. In this study, we selected PLS-SEM as an approach to tackle this issue. First, we developed a self-designed questionnaire to collect data related to international faculty's perspectives on their current working environment and their satisfaction. Second, in order to examine the causal relationships among international faculty perspectives, government perspectives, current work satisfaction, and overall work satisfaction, we proposed a PLS-SEM model with the hypotheses for testing. Finally, we verified the hypotheses with fitted indicators in PLS-SEM.

### 3.1. Research Framework

We focused on satisfaction as a factor influencing the recruitment and retention of international faculty. A conceptual framework was developed in the study based on a literature review that included the related constructs of government, institution management, and job satisfaction. According to Huang, both the government as well as the institutions should develop more strategies and efforts to ensure that international faculty members are satisfied with their working environment [45]. Figure 1 demonstrates the conceptual framework of this study. The causal relationships with the related variables are displayed, namely (a) institutional view (IV), (b) governance view (GV), (c) current work satisfaction (CWSat), and (d) overall satisfaction (OSat).

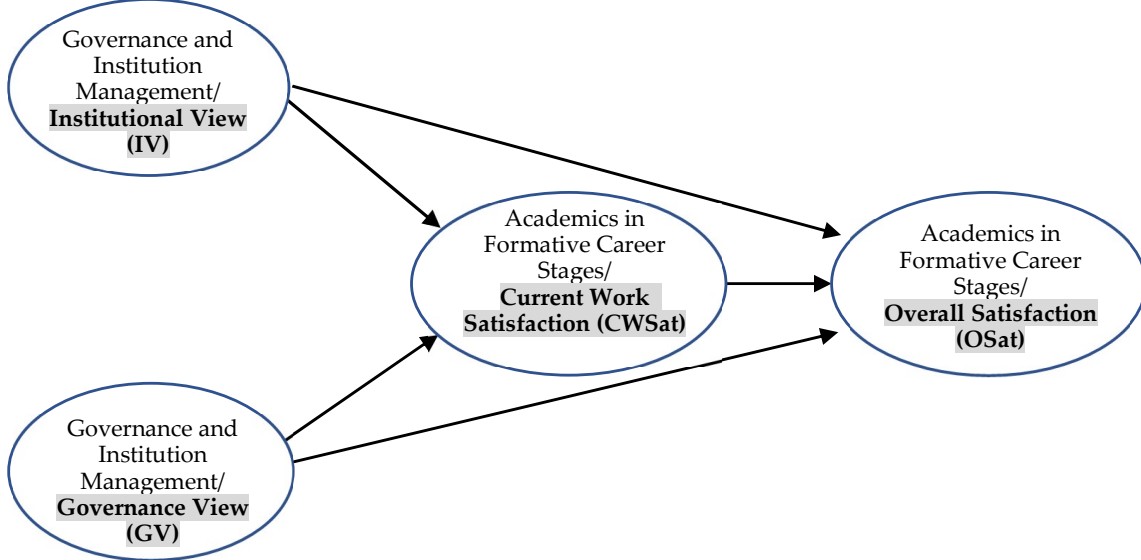

**Figure 1.** The conceptual framework of the study.

### 3.2. Instrument

Based on the research design, the research questionnaire can be classified into five domains. The content of the five domains is listed as follows:

a.  Career and Professional Situation (Question number A1 to A14), including academic rank, academic discipline, employment, the reason to teach or conduct research, teaching and research;

b.  Government and Institution Management (Question number B1 to B2), including government's policy for international faculty and how influential they are, personally, in helping to shape key academic policies at the institutional level;

c.   Academics in Formative Career Stages (Question number C1 to C8), including satisfaction with salary, job security, career opportunities, institutional prestige, personal independence in teaching and research;

d.   General Work Situation and Activities (Question number D1-D4), including satisfaction with employment situation, work situation, overall professional environment, and current job;

e.   Personal Background (Question number E1-E5), including gender, age, nationality, proficiency in Chinese, and institution type.

*3.3. Sampling*

To determine the target international faculty in the case of the higher education system, first, this study used the MOE database "A list of the academic expertise of university teachers in 2020" [46]. Second, based on the database, we sorted the names of potential international faculty members by university. Third, we removed the names that overlapped; for instance, one professor worked part-time at one university and full-time at another university. With a total of 88,976 academic teachers hired by Taiwan HEIs, we needed to perform data cleaning. Finally, data were sorted by faculty name with English names or not common last names to select potential foreigners in the database. Thus, with these potential international faculty names, these teachers' names had to be double-confirmed by checking the universities or departmental websites to know if they were international faculty. At the same time, it was important to gather international faculty's emails in order to send the invitation to participate in the survey.

Once the international faculty who were currently employed by Taiwan HEIs had been selected, the study was conducted using an online questionnaire platform to distribute the online survey. In the beginning, around 256 targeted international faculty members were sent the online questionnaires. This study belonged to probability sampling, and all the participants were treated equally. By way of the online survey platform, we successfully collected data. Finally, the total number of data contained 89 participants, but 9 of them had invalid or incomplete questionnaires. Overall, 80 questionnaires appeared to be valid.

*3.4. Fitted Samples for PLS-SEM*

PLS-SEM can include as many indicators and/or path relationships as necessary to meet the sample size requirements for each sub-model. In order to evaluate hypothesized effects, researchers need to carefully consider the most appropriate sampling strategy for their population as well as the sample size required for proper power calculations [4]. Conducting PLS-SEM, previous studies have suggested some useful guidelines that should be followed. For example, to determine the appropriate sample size, Kock and Hadaya recommended incorporating the model's background characteristics, the distributional properties of the data, the psychometric properties of the variables, and the degree of the relationship between the variables [47]; Hair et al. proposed the minimum R-squared method for estimation of the minimum sample size. They noted that four criteria can be examined in a structural equation model to determine sample size [48]:

a.   The significance level;

b.   The statistical power;

c.   The minimum coefficient of determination ($R^2$ values) used in the model;

d.   The maximum number of arrows pointing at a latent variable.

Cohen argued that sample size requirements should be based on three criteria: First, the minimum $R^2$ refers to the number of arrows pointing at a latent variable [49,50]. Second, the significance level is taken into account. Third, the minimum $R^2$ has to be considered for the model. To estimate the minimum sample size, the minimum $R^2$ in the model is commonly used. Hair et al. provided sample size recommendations for a power of 80% [51]. Based on the suggestion by Marcoulides and Saunders, the sample size in typical marketing research should have a significance level of 5%, a statistical power of 80%, and $R^2$ values of at least 0.25 [52]. Nevertheless, there is no clear literature that shows the suggested sample

size in educational research. Hair et al. suggested a minimum sample size of 52 to meet the significance level of 0.5 which has statistical power of 80% in the study [51]. The 80 samples in this study fit the requirement for conducting PLS-SEM.

### 3.5. Statistical Analysis

First, this study used SPSS (a statistical software program of IBM) to conduct descriptive statistical analyses to fulfill the first research purpose. A descriptive statistic is crucial in the data cleaning process, particularly when they are quantitatively describing or summarizing the details in a set of information. Second, we conducted PLS-SEM to verify the proposed model focusing on "Institution Management Perspectives", "Government Management Perspectives", "Academics in Formative Career Stages about Current Work Satisfaction", and "General Work Situation and Activities with Overall Satisfaction". The PLS-SEM was reviewed and the causal relationships were evaluated to fulfill the second research purpose.

PLS-SEM can be viewed as a nonparametric algorithm computation to determine latent variable scores [53]. Specifically, PLS-SEM uses composites as inputs and runs regressions with the aim of maximizing the explained variance of the endogenous constructs [53]; PLS-SEM can also be used to evaluate a theory from a predictive perspective [4,54]. Several advantages accompany the composite-based nature of the PLS algorithm, including its ability to predict out of sample and to employ composite scores for additional analyses [4,54]. PLS-SEM is more flexible generally, as the models are less constrained in terms of identification. In this way, formative measurement models can be utilized more effectively, and convergence can be ensured more easily [54]. PLS-SEM helps to avoid the problem of factor indeterminacy, which can occur when factor-based SEM provides determinate composite scores [3]. Sarstedt and Mooi pointed out that PLS-SEM does not assume residual distributions [55]. Therefore, researchers who employed the nonparametric bootstrapping procedure examined the confidence interval and tested the parameter significance [54]. In this study, PLS-SEM was used to transform the information of data distribution, the measured construct, discriminant validity, and the structural causal relationship. In regards to the survey instrument, reliability was used to determine whether the items in the study measure the same construct. Based on the suggestion of previous studies, composite reliability (CR) was considered to determine internal consistency. A CR value > 0.7 is required for it to be deemed adequate [47,53,56], while Fornell and Larcker indicated that the reliability statistic greater than 0.60 is considered a reliable indicator [57]. It provides a feasible threshold for selecting CR. In convergent validity analysis, the external factor loadings are greater than 0.5, and the average extracted variance (AVE) value is larger than 0.5. Hair et al. pointed out that these items represent good estimators with an outer loading larger than 0.5 [58]. Moreover, the heterotrait–monotrait (HTMT) ratio is an estimation of the correlations between the constructs. Hair et al. recommends using the HTMT criterion to assess discriminant validity [59]. Kline suggested a threshold of 0.85 or less [60], while Teo et al. recommended a liberal threshold of 0.90 or less [61].

## 4. Results

### 4.1. Descriptive Statistics for the Targeted International Faculty

The questionnaire featured 80 participants from international faculty members. Their average age was 50.32 years old for males and 47.90 years old for females. Males made up 73.8% and females 26.3% of the sample. Their nationalities were varied. The two main countries were Japan (25.0%) and the USA (17.5%). Except for the Americans and the Japanese, the international faculty currently working in Taiwan included individuals from Indonesia (6.3%), India (5.0%), Germany (5.0%), Australia (5.0%), France (3.8%), Korea (2.5%), Spain (2.5%), and Vietnam (2.5%). The other international faculty came from Austria, Brazil, the UK, Bulgaria, Canada, the Czech Republic, Greece, Iran, Israel, the Philippines, Poland, Russia, and South Africa. Furthermore, the level of Chinese proficiency of the international faculty in the areas of speaking, oral comprehension, reading, and writing

was an average of 3.3 out of 5, indicating the international faculty considered their level of Chinese proficiency to be mostly poor.

Most of the international faculty earned their degrees abroad. However, 66% of the international faculty earned post-doctoral degrees in Taiwan. There were 55.2% of the international faculty who have worked at a university or college outside Taiwan and 36.3% of the international faculty who have worked at a different university or college in Taiwan. In regards to their working conditions, most of the international faculty were employed at research-oriented public universities (40%). They were mostly lecturers or assistant professors (56.3%). Their academic discipline was mainly the humanities (37.5%); physical science and mathematics made up 11%. There were 83.8% of the international faculty hired full-time in their current positions. Furthermore, international faculty members were recruited by current institutions by applying directly to the institution (69.8%). The main reason why international faculty work in Taiwan is mainly for academic or professional purposes and fondness for Chinese life and culture.

Regarding the teaching conditions of international faculty, all of the content relating to teaching focuses on leading instruction, bachelor's degree courses, or equivalent courses. English is the language that is primarily used in teaching. The research part of this survey indicated that international faculty were engaged in research in the current academic year or the previous academic year. A total of 51.3% of international faculty worked independently without significant collaboration. Their main contributions were published articles, then written academic books or book chapters and papers presented at academic conferences. A relatively small percentage of 23.8% was what they had submitted or what they had co-authored. In addition, 22.5% of international faculty wrote discussion papers, reports, or monographs for funded projects. Another 12.5% were supervised doctoral dissertations. A little more than 8% were related to a patent or license secured over a process or invention and 6.3% to other scholarly contributions. Based on the analysis, the international faculty performed as is usual for the local faculty members.

### 4.2. Testing the Measurement Construct

The findings suggest that the CR values obtained for each construct range from 0.821 to 0.946, while Cronbach's alpha ranges from 0.682 to 0.919. Both CR and Cronbach's alpha are satisfactory and accepted, implying that the three latent constructs in this study have high levels of internal consistency according to Urbach and Ahlemann's criteria [56]. As a result of the analysis, some items were eliminated based on the AVE values for each construct which must be greater than 0.5 and the CR values greater than 0.7 [50,53]. This study also found that all items exceeded the specified level except items GV1, GV3, GV4, CWSat1, CWSat2, and CWSat 6–8 which had to be removed due to failure to meet the minimum requirements of factor loading. In spite of some items omitted, the findings indicate that the items in the study satisfied validity and reliability to measure all elements.

### 4.3. Discriminant Validity—HTMT

This study found that the HTMT ratios are below 0.90, and therefore discriminant validity can be established between two reflective constructs (see Table 1). It implies that the results support the existence of discriminant validity for every construct tested.

**Table 1.** Discriminant validity—HTMT.

| | Institutional View (IV) | Governance View (GV) | Current Work Satisfaction (CWSat) | Overall Satisfaction (OSat) |
|---|---|---|---|---|
| Institutional View (IV) | | | | |
| Governance View (GV) | 0.254 | | | |
| Current Work Satisfaction (CWSat) | 0.193 | 0.621 | | |
| Overall Satisfaction (OSat) | 0.194 | 0.367 | 0.600 | |

### 4.4. Verification of the Structural Model

In Table 2, the results of PLS-SEM reveal the *p* values that confirm if the path coefficients (β) are significant or non-significant. The results of hypotheses testing are displayed as follows:

**H1** evaluates whether international faculty's institutional view (IV) influences their current work satisfaction (CWSat). The result reveals that IV has no significant effect on CWSat ($\beta_1 = 0.102$, $p > 0.05$). Hence, H1 was not supported.

**H2** evaluates whether international faculty's governance view (GV) influences their current work satisfaction (CWSat). The result reveals that IV has a significant effect on CWSat ($\beta_2 = -0.463$, $p < 0.05$). As a result of the reverse questions, the path coefficient is negative, but the *p* value is still significant. Therefore, H2 was supported.

**H3** evaluates whether international faculty's current work satisfaction (CWSat) influences their overall satisfaction (OSat). The result reveals that CWSat has a significant effect on OSat ($\beta_3 = 0.514$, $p < 0.05$). Therefore, H3 was supported.

**H4** evaluates whether international faculty's institutional view (IV) influences their overall satisfaction (OSat). The result reveals that IV has no significant effect on OSat ($\beta_4 = 0.107$, $p > 0.05$). Therefore, H4 was not supported.

**H5** evaluates whether international faculty's governance view (GV) influences their overall satisfaction (OSat). The result reveals that GV has no significant effect on OSat ($\beta_5 = -0.088$, $p < 0.05$). Therefore, H5 was not supported.

**H6** evaluates whether international faculty's institutional view (IV), through current work satisfaction (CWSat), influences their overall satisfaction (OSat). The result reveals the indirect effect of IV -> CWSat -> OSat is 0.049 ($p > 0.05$). Therefore, H6 was not supported.

**H7** evaluates whether international faculty's governance view (GV), through current work satisfaction (CWSat), influences their overall satisfaction (OSat). The result reveals the indirect effect of GV -> CWSat -> OSat is -0.212 ($p < 0.05$). Therefore, H7 was supported.

**Table 2.** Summary of path analysis.

| Hypotheses | Structural Coefficient (β) | *p*-Values | Hypothesis Result |
|---|---|---|---|
| H1: IV -> CWSat | 0.102 | $p > 0.05$ | Not supported |
| H2: GV -> CWSat | −0.463 | $p < 0.05$ | Supported |
| H3: CWSat -> OSat | 0.514 | $p < 0.05$ | Supported |
| H4: IV -> OSat | 0.107 | $p > 0.05$ | Not supported |
| H5: GV -> OSat | −0.088 | $p > 0.05$ | Not supported |
| H6: IV -> CWSat -> OSat | 0.049 | $p > 0.05$ | Not supported |
| H7: GV -> CWSat -> OSat | −0.212 | $p < 0.05$ | Supported |

Figure 2 demonstrates the minimum R-squared method for minimum sample size estimation in this study. PLS-SEM tested the proposed model with four latent variables: institutional view (IV), governance view (GV), current work satisfaction (CWsat), and overall satisfaction (OSat). Figure 3 demonstrates the main impact of the SEM model with IV, GV, CWSat, and OSat. The critical path in the model is GV -> CWSat -> OSat. It shows that a mediation effect exists in the model.

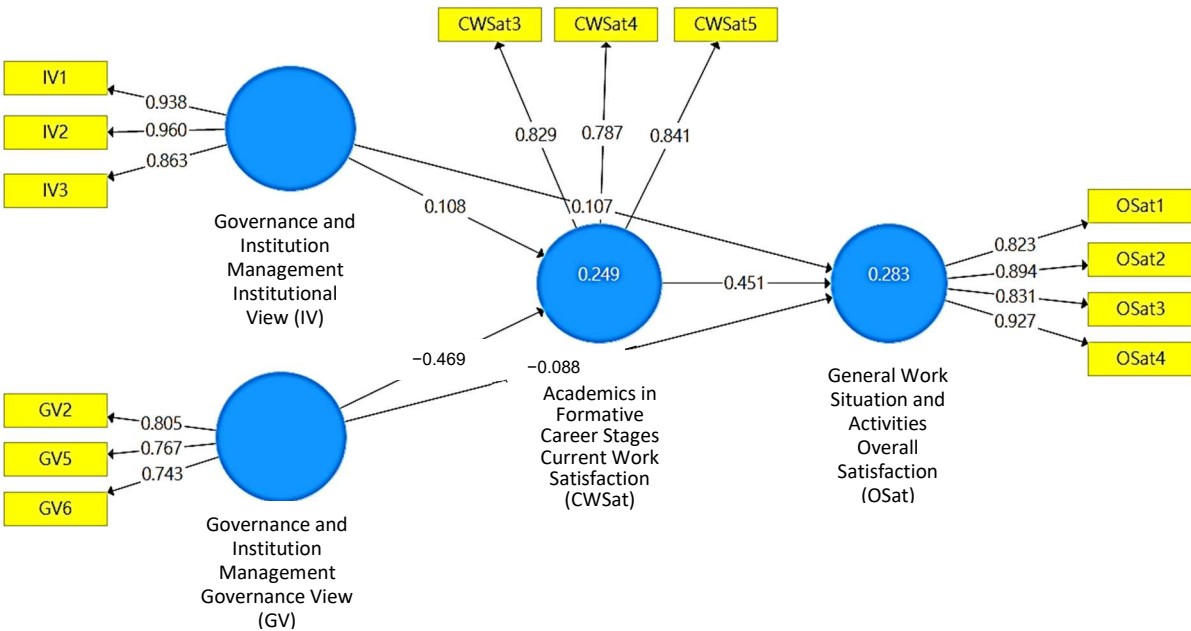

**Figure 2.** The minimum R-squared method for minimum sample size estimation in this study.

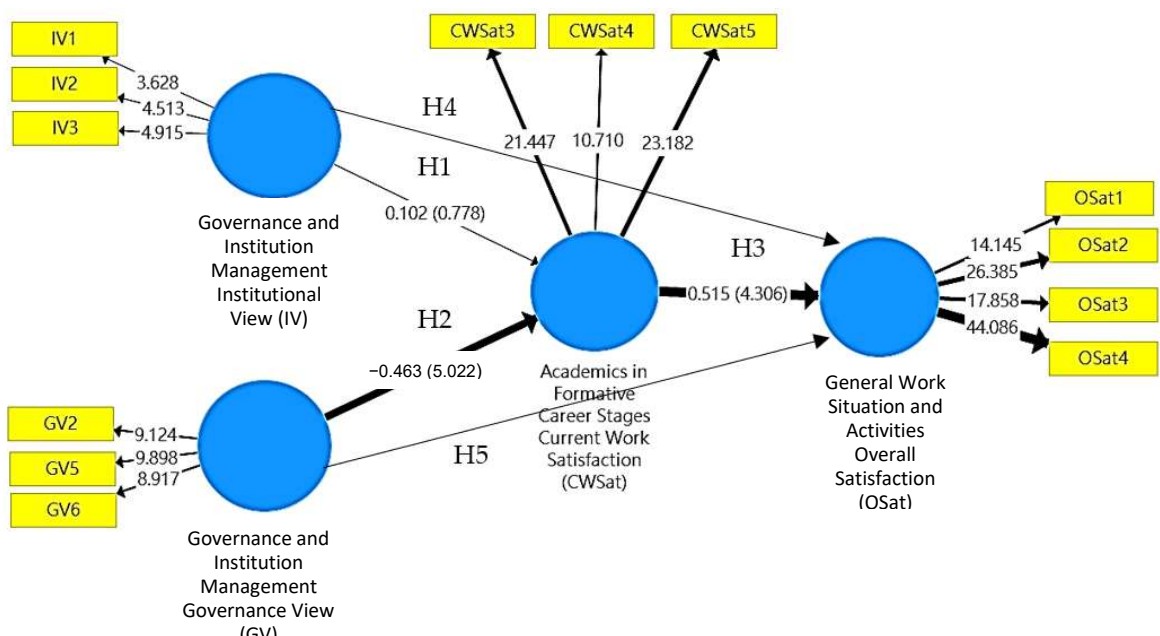

**Figure 3.** The main impact of SEM model with IV, GV, CWSat, and OSat.

## 5. Discussion

Higher education is increasingly recognizing its role in developing a sustainable society. However, the COVID-19 pandemic might have caused a pause in the implementation of some strategies [62,63]. The pandemic has impacted not only most citizens' lives but also the international faculty members' lives. This study demonstrated two parts of a survey of international faculty: one focused on the career and professional situation; the other focused on conceptual model testing. The design and findings can enrich the knowledge of this field. Firstly, the demographic profile survey may provide useful information for government or institutional policy makers. The international faculty in Taiwan maintains stable employment due to permanent employment or fixed-term employment with permanent positions. Taiwan has accepted a variety of nationalities of foreigners as indicated by the 80 respondents from 23 or more countries. Their academic

disciplines are primarily in the humanities and arts or engineering, manufacturing, and construction. This result can reflect the current academic trend in this specific country.

Secondly, the reason why international faculty members are choosing to teach or conduct research in Taiwan was addressed. According to IMD's latest report on world talent for 2021, Taiwan improved from 13th place in 2020 to 9th place in 2021 in the section on brain drain and migration of highly skilled foreign workers [64]. This indicates that the National Development Council (NDC) of Taiwan's goal to create an environment favorable for foreigners to work has been affected in this way by the Act for the Recruitment and Employment of Foreign Professionals [65]. Taiwan's higher education institutions provide a great academic environment for international faculty to devote themselves to their academic and professional pursuits. For policy makers, the average international faculty growth was less than 1% in Taiwan's HEIs in recent years. Previous studies have used the Ministry of Education database or national surveys to determine the current distribution of international faculty [22]. The survey indicated that most international faculty members work full-time as lecturers or assistant professors in most public universities. Considering that the findings suggest that the governance view is negative in relation to academic satisfaction, it is critical that related policy makers consider supporting both public and private universities to recruit international faculty, such as by allocating more funding to support private higher education institutions.

Thirdly, satisfaction is an influential indicator to judge the effectiveness of the government's policy and institutional strategies. Previous studies have indicated that satisfaction can be considered with related factors [38–40,45,66]. This study found that the international faculty's governance view impacted their current work satisfaction, but institutional view did not (Hypothesis 1 and Hypothesis 2). The study also found that international faculty's current work satisfaction (CWSat) influences their overall satisfaction (OSat) (Hypothesis 3), while international faculty's institutional view (IV) and international faculty's governance view (GV) did not influence their overall satisfaction (OSat) (Hypothesis 4 and Hypothesis 5). The results indicate that the full model of SEM is not supported. The indirect effect only exists in Hypothesis 7, implying that the international faculty's governance view (GV), through current work satisfaction (CWSat), influences their overall satisfaction (OSat). From TQM's perspective, the research design and findings can simplify the phenomena and address the key point of the issue.

It is also possible to investigate the detailed reason, by way of qualitative approaches, behind their view of the working conditions in HEIs. For higher education internationalization purposes, the design of this study may provide helpful information on the condition of the international faculty that should be improved. The PLS-SEM can confirm that satisfaction is an influential indicator to evaluate the effect of recruiting and retaining international faculty for governmental or institutional policy makers. As previously mentioned, international faculty can accelerate the campus' internationalization that prompts to build sustainable higher education. The research approach can quickly address the core issue that provides specific and helpful information to ameliorate the issues of attracting and maintaining international faculty. However, the limited samples and subjective individual perceptions on the measurement indicators could be biased in the survey.

## 6. Implications

Since international campuses have become a movement in contemporary higher education settings, international faculty could be a unique target that needs to be addressed. Previous studies provided very limited literature on the unique group. This study addressed an example to explore this topic in higher education. First, we provided a case country with an online survey platform to collect the limited data. Second, we demonstrated why PLS-SEM can fit the target samples. Third, we demonstrated the logic of PLS-SEM to verify the proposed model. Even though the situations and experiences of international faculty are different between nations and cases, the integrated information is

useful. The findings can help policy makers or managers in higher education to enhance their institutional strategies to recruit and retain international faculty.

## 7. Conclusions

The international faculty may have a variety of cultural backgrounds, and their motivations for working could also vary. An adequate quantitative research design can help establish reasonable dimensions for specific research purposes. This study demonstrated how the instrument was designed and how PLS-SEM was used to tackle the international faculty issue. In the conceptual research framework, we found it is feasible to interpret the situations that international faculty may face in a specific country but not limited.

Considering that internationalization has become an important movement, related studies for policy purposes will emerge in higher education. This study, as an exploratory study using an innovative approach, provides an example for conducting further similar studies, not only regarding the theoretical framework but also regarding international faculty's working conditions in HEIs. In spite of the low sample size, this study showed that PLS-SEM can be conducted with limited samples to achieve validity and reliability. This study suggests that using PLS-SEM to interpret the perspective of international faculty can contribute to the further development or review of a theoretical framework to deeply understand the relationships among international faculty's perspectives on campus life. Since quality higher education is one of the targets of SDG 4, the findings may provide helpful information for considering the international faculty as an essential part of higher education sustainable development.

**Author Contributions:** Conceptualization, C.-C.C. and D.-F.C.; methodology, C.-C.C. and D.-F.C.; software, C.-C.C. and D.-F.C.; validation, C.-C.C. and D.-F.C.; formal analysis, C.-C.C. and D.-F.C.; investigation, C.-C.C. and D.-F.C.; resources, D.-F.C.; data curation, C.-C.C. and D.-F.C.; writing—original draft preparation, C.-C.C.; writing—review and editing, D.-F.C.; visualization, C.-C.C. and D.-F.C.; supervision, D.-F.C. All authors have read and agreed to the published version of the manuscript.

**Funding:** This research received no external funding.

**Institutional Review Board Statement:** Not applicable.

**Informed Consent Statement:** Not applicable.

**Data Availability Statement:** Most of the data transformation is contained within the article. Data are available on request due to restrictions. The data presented in this study are available on request from the corresponding author.

**Conflicts of Interest:** The authors declare no conflict of interest.

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
