# Peer review of "Exploring International Faculty’s Perspectives on Their Campus Life by PLS-SEM"

_sustainability, doi:10.3390/su14159340_

Round 1
Reviewer 1 Report
Thank you for the opportunity to review your submission.
(1) Why are the relationships between Institution View / Government View and the target construct OSat are missing –> this model suggests a full mediation. Please test the mediation empirically in PLS-SEM.
(2) Use recent recommendations on how to carry out a PLS-SEM analysis. For example, You need to assess discriminant validity using HTMT and the predictive power of the model using PLSpredict.
(3) With the mediator model, consider using an IPMA.
(4) Use the latest SmartPLS 4 which offers many improvements and new features for your analysis (its early access is free of charge).
Good luck with your research!
Author Response
Reviewer 1: Suggestions and responses
1. Why are the relationships between Institution View / Government View and the target construct OSat are missing –> this model suggests a full mediation. Please test the mediation empirically in PLS-SEM.
Response: We revised a full mediation model and extended the hypotheses H4, H5, H6, and H7. With added H4 and H5, the model has become a full mediation. The mediation effect tests are H6 and H7. The test model is displayed in
2. Use recent recommendations on how to carry out a PLS-SEM analysis. For example, you need to assess discriminant validity using HTMT and the predictive power of the model using PLSpredict.
Response: The discriminant validity with HTMT has been displayed in 4.3 and Table 1.
3. With the mediator model, consider using an IPMA.
Response: The mediation effect test has been added.
4. Use the latest SmartPLS 4 which offers many improvements and new features for your analysis (its early access is free of charge).
Response: Thanks for your helpful message. We have improved the results and created new features to fit PLS-SEM.

Reviewer 2 Report
The paper is generally well written and structured. The introduction is relevant and theory-based. Sufficient information about the previous study findings is presented for readers to follow the present study rationale and procedures.
However, in my opinion, the paper has some shortcomings:
(1) The second paragraph should be linked to the third paragraph. Where is International mobility coming from? International mobility should be appropriately linked to Universities across the globe.
(2) The introduction fails to present prevalent issues around the perspectives of international mobility of faculty across various countries. Kindly reveal the efforts of the government on the international mobility of faculty over the years with empirical insights and justifications.
(3) The author/s need to bridge existing theories in exciting ways, link work across disciplines, provide multi-level insights, and broaden the scope of international mobility among Faculty.
(4) How has this topic contributed to the realisation of Sustainable Development Goals (SDGs)? This needs to be clearly spelt out.
(5) Given these shortcomings, the manuscript requires minor revisions.
Methods and Analysis
(6) The methods were clearly spelt out but needed thorough justification. What is the unit of analysis? Why questionnaire and SEM? What was the source of the research instrument? How was the research instrument validated?
(7) Common method bias is needed to explain the network of causes and effects among latent variables in the model being studied.
(8) The hypotheses were appropriately analysed but lack adequate discussions of the findings
(9) Kindly highlight the major implications of the study and contributions to knowledge
GENERAL COMMENTS
- The paper has a similarity index of 19%. This is highly commendable.
- The use of current citation is commendable, but the author/s can intensify more efforts in this regard.
- This paper needs thorough editing.
In sum, this paper provides integration of literature, offers an integrated framework, provides value-added, and highlights directions for future inquiry.
Author Response
Reviewer’s suggestions and responses
The paper is generally well written and structured. The introduction is relevant and theory-based. Sufficient information about the previous study findings is presented for readers to follow the present study rationale and procedures.
However, in my opinion, the paper has some shortcomings:
- The second paragraph should be linked to the third paragraph. Where is international mobility coming from? International mobility should be appropriately linked to universities across the globe.
Response: We have moved the original second paragraph to a fitting place, see page 2-3. We also revised the first paragraph to make it more consistent.
- The introduction fails to present prevalent issues around the perspectives of international mobility of faculty across various countries. Kindly reveal the efforts of the government on the international mobility of faculty over the years with empirical insights and justifications.
Response: We revised the paragraph and added words: “In the global context, for example, the Korean governance launched “the Brain Pool Project” …; Japan has made greater efforts to attract international faculty, researchers, and talents by implementing several national-level policies in the 1990s. Attracting international faculty has become an increasingly important strategy for improving the quality and international competitiveness of national higher education systems in the OECD areas and many East Asian countries such as China and Singapore, and Malaysia……” See the second paragraph. (page 2)
In Taiwan, the Ministry of Education focused on selected higher education institutes possessing a reputation for high-quality research and launched “the Development Plan for World Class Universities and Research Centers of Excellence” in 2016. In the second stage (2011–2016), the project was renamed “the Aim for the Top University Plan”.
- The author/s need to bridge existing theories in exciting ways, link work across disciplines, provide multi-level insights, and broaden the scope of international mobility among faculty.
Response: Previous studies indicate that customer satisfaction is the crucial quality assurance component of TQM; for example, TQM practices are significantly and positively linked to customer satisfaction and service quality [42,43]. In this sense, the notion of TQM and practices can be extended to realize international faculty in higher education. Job and academic satisfaction could be useful indicators to reflect the related policies, strategies, and campus life.
- How has this topic contributed to the realization of Sustainable Development Goals (SDGs)? This needs to be clearly spelt out.
Response: We added some words to the conclusion section. “Since quality higher education is one of the targets in SDGs 4, the findings may provide helpful information to thinking the international faculty as an essential part of the higher education sustainable development.”
- Given these shortcomings, the manuscript requires minor revisions.
Response: We have revised the (1) to (4) as the suggestions.
Methods and Analysis
- The methods were clearly spelt out but needed thorough justification. What is the unit of analysis? Why questionnaire and SEM? What was the source of the research instrument? How was the research instrument validated?
Response: We have reorganized the methods and analysis section to make it more clear and more coherent.
- Common method bias is needed to explain the network of causes and effects among latent variables in the model being studied.
Response: We added the following explanation: “While the limited samples and subjective individual perception on the measurement indicators could be biased in the survey.” (page 18)
- The hypotheses were appropriately analyzed but lacked adequate discussion of the findings
Response: We added the following discussion: “This study found international faculty’s governance view impacted on their current work satisfaction, but the institutional view did not (Hypothesis 1 and Hypothesis 2). The study also found international faculty’s current work satisfaction (CWSat) influences their overall satisfaction (OSat). (Hypothesis 3), while international faculty’s institutional view (IV) and international faculty’s governance view (GV) did not influence their overall satisfaction (OSat) (Hypothesis 4 and Hypothesis 5). The results indicate that the full model of SEM is not supported. The indirect effect only exists in Hypothesis 7 that international faculty’s governance view (GV), through current work satisfaction (CWSat), influences their overall satisfaction (OSat).” (page 18-19)
- Kindly highlight the major implications of the study and contributions to knowledge
Response: We have added the implications section to highlight the major implications.
GENERAL COMMENTS
- The paper has a similarity index of 19%. This is highly commendable.
Response: We have revised the text from some of the citations to reduce the similarity.
- The use of the current citation is commendable, but the author/s can intensify more efforts in this regard.
Response: We have updated some references, for example [41], [42], [43]
- This paper needs thorough editing.
Response: We have checked through the paper. The tracking system has shown the revision.
In sum, this paper provides integration of literature, offers an integrated framework, provides value-added, and highlights directions for future inquiry.

Reviewer 3 Report
“The purpose of this study is to explore the perspectives of international faculty members on their life in higher education institutes by using partial least square structural equation modeling (PLS-SEM)” (Please separate the objective from the statistical techniques, it is better to mention this technique in the methodology part)
Please include the sampling technique in the abstract.
You have used future tense (will) in the abstract to indicate your objectives !!!! Please make it present perfect tense. (Such as the design has examined the relationships among working conditions, views of institutions, view of 19 government measures, and levels of satisfaction through PLS-SEM)
I did not find the major findings of your study in the abstract part.
The introduction part is well-organized.
In the literature review part, you should first clarify the basic concepts of International faculty; whatever you have already done in the second part of the literature, you should make it first in the literature review part.
In the literature review part (2.3 Job satisfaction and academic satisfaction), you are requested to follow the following article to get the proper idea.
Work to family, family to work conflicts and work family balance as predictors of job satisfaction of Malaysian academic community (https://www.emerald.com/insight/content/doi/10.1108/JEC-05-2020-0098/full/html)
Research framework and hypotheses development must be after the literature review part. It should not be included in the methodology part. You need to shift the 3.1 part after the literature review part.
In the sample (3.3) part, you have focused on determining the sample size. You must need focus on the sampling technique. You may follow the following article to get an idea about sampling !!!
Sampling Techniques (Probability) for Quantitative Social Science Researchers: A Conceptual Guidelines with Examples (https://sciendo.com/article/10.2478/seeur-2022-0023)
The analysis part is quite ok.
The discussion part is well organized as it has been designed based on the findings of the study.
Before going to the conclusion, it is must to put the implication part.
Thanks.
Author Response
Reviewer 2: suggestions and responses
1. “The purpose of this study is to explore the perspectives of international faculty members on their life in higher education institutes by using partial least square structural equation modeling (PLS-SEM)” (Please separate the objective from the statistical techniques, it is better to mention this technique in the methodology part)
Response: The related analyses have been separated to fulfill the first and second research purposes, see the revision on p.7 and p.8. We also reorganized the “statistical analysis” in the method section.
2. Please include the sampling technique in the abstract.
Response: We added the text “It is about 31.25% of the responses from the online mailing survey technique” in the abstract.
3. You have used future tense (will) in the abstract to indicate your objectives! Please make it present perfect tense. (Such as the design has examined the relationships among working conditions, views of institutions, view of 19 government measures, and levels of satisfaction through PLS-SEM)
Response: We have revised the tense.
4. I did not find the major findings of your study in the abstract part.
Response: We have added the information on the mediation effect: “The result of PLS-SEM suggests that the international faculty perspective of government, through their current work satisfaction, impacts overall satisfaction. This study has confirmed the mediation effect in the testing model”.
5. The introduction part is well-organized.
Response: Thanks.
6. In the literature review part, you should first clarify the basic concepts of International faculty; whatever you have already done in the second part of the literature, you should make it first in the literature review part.
Response: We have moved the “definitions of international faculty” to 2.1 and adjusted the sequences of the other subsections. See p. 3
7. In the literature review part (2.3 Job satisfaction and academic satisfaction), you are requested to follow the following article to get the proper idea. Work to family, family to work conflicts and work family balance as predictors of job satisfaction of Malaysian academic community (https://www.emerald.com/insight/content/doi/10.1108/JEC-05-2020-0098/full/html)
Response: We have added the following important reference in the text to support our argument. See p. 5
“Rahman, M.M.; Ali, N.A.; Jantan, A.H.; Mansor, Z.D.; Rahaman, M.S. Work to family, family to work conflicts and work family balance as predictors of job satisfaction of Malaysian academic community. J. Enterp. Communities. 2020, 14, 4, 621-642.”
8. Research framework and hypotheses development must be after the literature review part. It should not be included in the methodology part. You need to shift the 3.1 part after the literature review part.
Response: We have moved the hypotheses to Section 2. See p.5
9. In the sample (3.3) part, you have focused on determining the sample size. You must need focus on the sampling technique. You may follow the following article to get an idea about sampling! Sampling Techniques (Probability) for Quantitative Social Science Researchers: A Conceptual Guidelines with Examples (https://sciendo.com/article/10.2478/seeur-2022-0023)
Response: We have added the sampling technique information in the text, see p. 7, for example, “It is about 31.25% of the responses from the online survey technique. We consider the population has an equal chance of being chosen, it belongs to a probability sampling technique.”
10. The analysis part is quite ok.
Response: Thanks.
11. The discussion part is well organized as it has been designed based on the findings of the study.
Response: Thanks.
12. Before going to the conclusion, it is must to put the implication part.
Response: The implication part has been added on p. 12
